# X-ray crystallographic and hydrogen deuterium exchange studies confirm alternate kinetic models for homolog insulin monomers

Esra Ayan[1,2,3*], Miray Türk[4], Özge Tatlı[5,6], Sevginur Bostan[3,7], Elek Telek[8], Baran Dingiloğlu[4], B. Züleyha Doğan[3], Muhammed Ikbal Alp[3*], Ahmet Katı[2], Gizem Dinler-Doğanay[4,9], Hasan Demirci[1,10]

1 Department of Molecular Biology and Genetics, Faculty of Science, Koç University, Istanbul, Türkiye, 2 Experimental Medicine Research and Application Center, University of Health Sciences, Istanbul, Türkiye, 3 Research Institute for Health Sciences and Technologies (SABITA), Neuroscience Research Center, Istanbul Medipol University, Istanbul, Türkiye, 4 Department of Molecular Biology-Genetics and Biotechnology, Istanbul Technical University, Istanbul, Türkiye, 5 Division of Translational Cancer Research, Department of Laboratory Medicine, Lund University, Lund, Sweden, 6 Department of Molecular Biology and Genetics, Istanbul Medeniyet University, Istanbul, Türkiye, 7 Department of Physiology, International School of Medicine, Istanbul Medipol University, Istanbul, Türkiye, 8 Department of Biophysics, Medical School, University of Pécs, Pécs, Hungary, 9 Department of Molecular Biology and Genetics, Faculty of Science and Letters, Istanbul Technical University, Istanbul, Türkiye, 10 Stanford PULSE Institute, SLAC National Laboratory, Menlo Park, California, United States of America

* esraayan20@ku.edu.tr (EA); malp@medipol.edu.tr (MIA)

## Abstract

Despite the crucial role of various insulin analogs in achieving satisfactory glycemic control, a comprehensive understanding of their in-solution dynamic mechanisms still holds the potential to further optimize rapid insulin analogs, thus significantly improving the well-being of individuals with Type 1 Diabetes. Here, we employed hydrogen-deuterium exchange mass spectrometry to decipher the molecular dynamics of newly modified and functional insulin analog. A comparative analysis of H/D dynamics demonstrated that the modified insulin exchanges deuterium atoms faster and more extensively than the intact insulin aspart. Additionally, we present new insights derived from our 2.5 Å resolution X-ray crystal structure of modified hexamer insulin analog at ambient temperature. Furthermore, we obtained a distinctive side-chain conformation of the Asn3 residue on the B chain (AsnB3) by operating a comparative analysis with a previously available cryogenic rapid-acting insulin structure (PDB_ID: 4GBN). The experimental conclusions have demonstrated compatibility with modified insulin's distinct cellular activity, comparably to aspart. Additionally, the hybrid structural approach combined with computational analysis employed in this study provides novel insight into the structural dynamics of newly modified and functional insulin vs insulin aspart monomeric entities. It allows further molecular understanding of intermolecular interrelations driving dissociation kinetics and, therefore, a fast action mechanism.

**Data availability statement:** Data is available from https://www.rcsb.org/structure/8Z4B or within the manuscript.

**Funding:** E.T. was supported by the University of Pécs Medical School, a grant from Dr. Szolcsányi János Research Fund (KA-2022-09), as well as the grant of Dr. Romhányi György fellowship for young scientists (ÁOK-IK). This research was financially supported by The Scientific and Technological Research Council of Turkey (TUBITAK) (Project no: 122R061). E.A. was supported by received funding from the TUBITAK 2244 Program (Project no: 119C132). Additionally, M.T. and B.D. received funding from the TUBITAK 2211 Program. The funders had no role in study design, data collection and analysis, decision to publish, or preparation of the manuscript.

**Competing interests:** The authors have declared that no competing interests exist.

# Introduction

Insulin plays a critical role in regulating cellular glucose uptake. Deficiency or insensitivity to insulin leads to diabetes mellitus, a prevalent and severe metabolic disorder. Treatment typically involves administering insulin and derivatives, which are integral interventions [1]. Insulin is primarily hexameric (consist of six identical monomer ~36 KDa) and dimeric (consist of two identical monomer ~12 KDa) during production, delivery, and circulation. However, upon binding to its receptor, the protein adopts a monomeric conformation (~5 KDa) [2], which differs from its solution structure observed in NMR and X-ray crystallographic studies [3,4]. Environmental factors and sequence modifications that favor the monomeric state enhance fibrillation [5] and degradation [6], compromising the stability of therapeutic formulations. Consequently, there is significant interest in medicine, biophysics, and structural biology in understanding the structural configurations of the insulin monomer under physiological conditions.

Structural analyses of the fully active insulin monomer [7] largely favor the hexameric form's T-state identified in crystallographic studies [8]. Research studying amide hydrogen exchange for lispro analog (the active component of Humalog®; Eli Lilly and Co.) present as a monomer in solution reveals limited protection at the A-chain N-terminus, the B-chain N-terminus, and the B-chain C-terminus [9]. The observed exchange at relevant sites suggests that these regions are solvent-exposed, though the exact nature of these states remains incompletely understood. Experimentally studying the structural ensemble of the wild-type monomer is challenging due to its tendency to undergo fibrillation and oligomerization at concentrations ranging from micromolar to millimolar [10]. Experimental studies of insulin monomers often focus on sequences with substitutions or conditions significantly deviating from physiological parameters (e.g., pH > 8), which may influence the degree of structural disorder [11,12]. Comprehensive all-atom molecular dynamics simulations are needed to investigate the insulin monomer in greater detail. Such simulations provide a direct method for exploring microscopic structures and the fundamental forces that stabilize them. Unbiased nanosecond-scale simulations were performed on the porcine insulin monomer, starting from the T state observed in the hexameric crystal structure [8]. These simulations revealed disordered regions in the N- and C-termini of the B-chain, while the AN-helix remained stable. The root-mean-square deviation from the T state indicated significant unfolding, though the structural details were limited. Additionally, bias-exchange metadynamics simulations at low pH and high temperature showed a diverse ensemble dominated by fully unfolded states [13]. The structural characterization of a protein containing disordered regions poses experimental difficulties due to the rapid interconversion of conformations within picosecond to millisecond lifetimes [14]. While hydrogen/deuterium exchange mass spectrometry (HDX/MS) represents a valuable approach for gaining insights into protein conformation and disordered structural dynamics [15,16], previous studies have focused on conducting H/D reactions on oligomeric insulins rather than monomeric conformer [17,18].

This study attempted to elucidate the underlying mechanism governing the dynamic changes in our monomer of less-stable insulin analog (RB22K-PB28D, hereafter referred to as INSv) compared to its counterpart, insulin aspart (PB28D, hereafter referred to as IAsp). This was achieved by introducing a mutation in the 22nd residue on the B chain, replacing arginine with a more flexible lysine residue together with a substitution at the 28th residue on chain B from proline to aspartic acid, and investigating its impact on protein stability and electrostatic interactions. The model system employed for this study was IAsp [19]. The ArgB22 residue in insulin was specifically mutated to Lys (RB22K) to address both biochemical and industrial challenges. This mutation reduces the formation of undesired cleavage products, as trypsin

has a higher specificity for Arg than Lys [20], thereby improving production consistency and yield. Additionally, replacing Arg with Lys lowers the isoelectric point (pI) of the insulin molecule, promoting the dissociation of the inactive hexameric form into the biologically active monomer under physiological conditions [21]. Lys also introduces greater flexibility to the insulin structure due to its shorter side chain, which can influence the stability and receptor-binding dynamics of the protein [22]. Importantly, this mutation eliminates the need for additional enzymes, such as carboxypeptidase [23], allowing insulin to be processed using trypsin alone, streamlining production and reducing costs. The RB22K mutation not only addresses industrial optimization goals but also provides a valuable model for studying the structural and functional implications of modifications at the functionally critical B22 position.

Once RB22 to KB22 substitution over IAsp, the activity of the INSv monomer in the cells has been initially confirmed over calcium imaging, serving the IAsp monomer as a control. Subsequently, the stability of the INSv monomer was assessed using the differential scanning calorimetry (DSC) technique, comparing with IAsp monomer stability. Hydrogen-deuterium exchange mass spectrometry (HDX-MS) analysis explored these monomers' intrinsic and distinct dynamics in solution. The findings from this experimental approach were further validated computationally by employing Gaussian Network Model (GNM) analysis. Finally, a 2.5 Å resolution X-ray crystal structure of its hexamer was determined through the newly developed multi-well X-ray crystallography technique of Turkish Light Source conducted at ambient temperature, elucidating the conformational differences between the hexamers of both INSv and IAsp.

## Methods

### Production of recombinant double mutant INSv

The recombinant INSv gene was cloned to the pET28a (+) vector and was transformed into the *E. coli* BL21 Rosetta-2 strain for overexpression. Bacterial cell cultures, each containing the recombinant INSv protein genes, were cultivated independently in 6 liters of either standard LB-Miller media or Terrific Broth (TB), supplemented with the final concentration of 35 μg/mL chloramphenicol and 50 μg/mL kanamycin at 37°C. Cultures were agitated by using a New Brunswick Innova 4430R shaker at 110 rpm until they reached an optical density at 600 nm ($OD_{600}$) of approximately 0.7 for each culture. Recombinant protein expression was induced with Isopropyl b-D-1-thiogalactopyranoside (IPTG) at a final concentration of 0.4 mM. The incubation for protein overexpression was conducted at 37°C for 4 hours. Subsequently, cells were harvested at 4°C using a Beckman Allegra 15R desktop centrifuge at 3500 rpm for 20 minutes. Verification of protein expression was conducted through precast TGX-mini protean gradient SDS-PAGE from BioRad.

### The solubilization process of INSv

For the partial purification and complete solubilization of inclusion bodies, cell pellets (1 gram) containing INSv proteins were individually resuspended in 10 mL of lysis buffer containing 25 mM Tris-HCl, pH 8.0, 5 mM Ethylenediaminetetraacetic acid (EDTA) and subsequently homogenized. Bacterial cells were lysed via sonication, and the resulting cell debris underwent low-speed centrifugation (6000 × g, 4°C, 7 minutes). The obtained pellets were subjected to suspension in wash buffer A containing 25 mM Tris-HCl, pH 8.0, 5 mM EDTA supplemented with 0.1% (v/v) Triton X-100 and 1 M urea, followed by wash buffer B containing 25 mM Tris and 2 M urea. The suspension was sonicated for 30 seconds in an ice bath and then centrifuged (8000 × g, 4°C, 20 minutes). To further isolate inclusion bodies, the debris

was resuspended at 0.1 g/mL in binding buffer containing 25 mM Tris-HCl, pH 8.4, and 5 mM 2- mercaptoethanol (BME) for INSv, which contained 8 M urea. Centrifugation at $10,000 \times g$ for 30 minutes at 4°C allowed recovery of inclusion bodies containing the fusion proteins, and the resulting supernatants were filtered through a 0.2-micron Millipore filter.

## Purification, refolding, and tryptic digestion process of INSv

Sulfitolysis of INSv was conducted at 25°C for 4 hours using 200 mM sodium sulfite ($Na_2SO_3$) and 20 mM sodium tetrathionate ($Na_2S_4O_6$). Subsequently, the manual gravity Ni-NTA affinity chromatography was employed, involving column equilibration with one column volume of a buffer containing 25 mM Tris-HCl, pH 8.0, 5 mM BME, and 8 M urea and elution with a buffer containing 25 mM Tris-HCl, pH 8.0, 5 mM BME, 8 M urea, and 250 mM imidazole. The protein sample purity was assessed at each stage through 20% SDS-PAGE gel analysis and Coomassie staining. High-purity fractions were collected and filtered through a 0.2-micron Millipore filter. The INSv refolding process is completed by using standard published protocols (Khosravi et al., 2022; Kim et al., 2015; Min et al., 2011). Purified proinsulin (0.5 mg/mL) underwent dialysis against 25 mM Tris-HCl pH 9.00 with 8 M urea and 0.5 mM EDTA at 4°C. After centrifugation ($10.000 \times g$, 4°C, 30 minutes) to remove potential aggregates, the resulting mixture was dialyzed in a refolding solution with 0.1 M Gly/NaOH at pH 10.5, 0.5 M urea, 0.5 mM EDTA, 5% glycerol, and stirred at 15°C overnight. Refolded INSv was further dialyzed against 25 mM Tris-HCl, pH 8.0, with 5% glycerol for 18 hours.

Mature insulin is derived from proinsulin, a precursor molecule cleaved by proteolytic enzymes such as trypsin. Proinsulin consists of three leading peptide chains: the B-chain, C-peptide, and A-chain. The C-peptide is crucial in adequately folding proinsulin. Following this folding process, the C-peptide must be removed by proteases like trypsin to activate insulin [24–26]. Enzymatic digestion can lead to impurities, including A21 desamido insulin, B30 des-threonine insulin, insulin ethyl ester, and arginine- and diarginine-insulin. The primary contaminants of concern are A21 desamido insulin and B30 des-threonine insulin [27]. To reduce the production of B30 des-threonine insulin, regulating the reaction temperature and the ratio of trypsin to CPB [25]. Therefore, proinsulin underwent tryptic cleavage using well-optimized trypsin 1:1000 (v/v) ratio for 4 hours at 30°C. Following the proteolytic cleavage, pH precipitation was employed by adding 20 mM citric acid to the protein sample.

Upon precipitation, INSv was centrifuged at 3500 rpm at 4°C for 5 minutes, and the resulting pellet was dissolved in 20 mM citric acid to quench trypsin activity and subsequent removal of trypsin. After solubilization, INSv was filtered through a 0.2 μM pore diameter filter and subjected to size exclusion chromatography (SEC) using a 20 mM citric acid buffer. High-purity fractions were collected, and pH precipitation was performed using 1 M Tris, pH 10.0. The INSv pellet was then solubilized using 25 mM Tris-HCl, pH 9.0 buffer, and the concentration was adjusted to 1000 μg/mL for HDX-MS analysis.

## Monomerization of insulin aspart (IAsp)

Precipitation through pH adjustment and subsequent solubilization in 25 mM Tris-HCl pH 9.0 buffer were employed to convert IAsp (NovoRapid) from a hexameric to a monomeric state [11,12,27]. For further buffer exchange and concentration of INSv monomer, a 3K amicon ultrafiltration unit was used ten times washing with 25 mM Tris-HCl pH 9.0 buffer. The resulting monomeric form of NovoRapid was then adjusted to a 1000 μg/mL concentration for HDX-MS analysis.

## Circular dichroism analysis

The far-UV CD spectrum (195–250 nm) of INSv samples was recorded using a Jasco J-1500 circular dichroism spectrometer (Jasco Co., Ltd., Tokyo, Japan), employing 1 mm quartz cells, a 2 nm bandwidth, a 0.2 nm resolution, and a 50 nm/min scanning speed. The samples were prepared in three distinct pH buffers (pH 3.0, pH 9.5, pH 10.0). All analyses were conducted in triplicate. The CD spectrum was analyzed using Spectra Analysis (Jasco) software, and the data were smoothed utilizing a Savitzky-Golay filter.

## INSv and IAsp-induced in-vitro activity experiments

**Animals.** Animal procedures were performed in accordance with the European Community directive 86/609/CEE and supervised by the Committee on Ethics of Animal Experimentation of Istanbul Medipol University (IMUHADYEK). Mouse pups aged 0–1 days were provided from C57Bl/6j mice. Breeders provided food and water to animals as ad libitum, and housing of animals was provided in a 12-h light/dark cycle in stable conditions of temperature (22 C) and humidity (60%). Euthanasia of mouse pups was achieved by decapitation using large scissors. No need to use anesthesia due to providing decapitation of mouse pups directly using big scissors.

**Primary hippocampal cell culture.** Newborn mice aged 0–1 days were decapitated directly using big scissors, following ethical guidelines. Their brains were extracted in a sterile laminar flow cabinet and immediately immersed in a Hibernate-A (Gibco-A1247501) solution, including 1% Antibiotic and 1% Glutamax. The hippocampi were carefully dissected under a stereomicroscope and placed in an enzymatic solution containing L-15 (Sigma L5520), 1% antibiotics, 1% Glutamax, 2% B27, 1% papain, and 1% DNAse, and maintained at 4°C for 45 minutes. Once performing the enzymatic treatment, the hippocampal tissues were mechanically triturated using a series of three glass Pasteur pipettes with sequentially decreasing tip diameters. The resulting cellular suspension was then transferred to an enzyme inhibition medium containing L-15 solution, 1% antibiotics, 1% Glutamax, 2% B27, and 10% FBS and incubated at 4°C for 15 minutes. The cells suspended in the enzyme inhibition medium were centrifuged at 1000 rpm for 5 minutes. The cells were resuspended with Neurobasal-A containing 1% antibiotics, 1% Glutamax, and 2% B27 and seeded into cell culture dishes at a density of 15,000–20,000 cells/mm². After the initial seeding, the cells were incubated at 37°C in an atmosphere containing 5% $CO_2$ for 2 hours. To maintain optimal conditions, the total volume in the culture dish was adjusted to 1500 μL with additional maintenance medium. GCaMP plasmid (Addgene plasmid #100843) [28] was transduced to the cells using an adeno-associated virus (AAV) known for its efficacy in neuronal systems. Half of the medium was replenished every three days. Experimental procedures were initiated on day 21, after a significant increase in GCaMP expression and following the maturation of the neuronal network.

**Calcium imaging of INSv and IAsp-induced activities.** Sequential action potentials trigger a substantial influx of calcium ions into the cell. Upon entry, these calcium ions interact with the binding domains of the GCaMP protein, inducing conformational alterations within the protein structure. These structural changes activate the green fluorescent protein (EGFP) linked to GCaMP, enhancing fluorescence intensity. To visualize this fluorescence, the EGFP must be excited using blue light at a wavelength of 488 nm, leading to the emission of fluorescent green light at a wavelength of approximately 515 nm. The calcium activities within hippocampal neuronal networks were monitored and visualized using a fluorescent microscope equipped with a connected camera. Imaging was conducted using 10x objectives. Images were captured at a frequency of 1 frame per second and subsequently compiled

into video recordings. The video recordings were analyzed using Zen Blue (Zeiss) software, wherein selected cell bodies were designated as regions of interest (ROIs). The average light intensity variation within these areas was calculated for every frame. Calcium transient intensities were determined using the formula $\Delta F/F0$, where $\Delta F$ represents the difference between the fluorescence intensity (F) in the region of interest and the baseline intensity (F0). The average light intensity variation within these areas was calculated for each frame. The results were further processed and analyzed using Excel (Microsoft) and Prism 9 (GraphPad Software, La Jolla, CA, USA). Temporal variations in light intensity observed within the selected cell bodies were graphically represented over time. The quantification of active cells was conducted through manual enumeration.

**Ethics declarations.** All authors confirmed that all experimental protocols were approved by Committee on Ethics of Animal Experimentation of Istanbul Medipol University (IMUHADYEK). All authors confirmed that all methods were carried out in accordance with European Community directive 86/609/CEE. All author confirmed that all methods are reported in accordance with ARRIVE guidelines.

## Kinetics of insulin monomers defined through HDX-MS analysis

HDX-MS analysis has been performed at five different time points ranging from 12 s to 24 h. Purified INSv and IAsp were diluted in the equilibration buffer containing 50 mM Tris-HCl; pH 9.5. For labeling, 2.5 µl ($\geq$ 35 pmol) of each sample was incubated with 37.5 µl labeling buffer containing 50 mM Tris-HCl in D2O; pD 9.5; pH 9.1 at room temperature for 0, 0.2, 2-, 20-, 200-, or 1440-mins incubation periods. Labeled samples were quenched with 40 µl pre-chilled quench buffer containing 50 mM Tris-HCl, 3M guanidine hydrochloride (GnHCl), 100 mM tris (2-carboxyethyl) phosphine (TCEP); pH 2.3, then the sample was incubated on ice for one minute to inhibit deuterium back-exchange. All labeling time points were subject to triplicate analyses. $t_0$ samples were used as reference and prepared as quadruplate. Deuterated samples were digested using The Enzymate BEH Pepsin Column (Waters Corp., Manchester, UK) at 20°C. Upon 3 minutes of transferring digested peptides at Waters BEH C18 VanGuard trapping pre-column (flow rate 70 µl/min in 0.2% formic acid), proteins underwent online pepsin digestion and liquid chromatography/mass spectrometry (LC/MS) using the nanoACQUITY UPLC system and the Synapt G2-Si Q-Tof mass spectrometer from Waters Corp. (Milford, MA, USA). Peptide separation conditions included a 6-minute gradient with 0.2% formic acid, 35% acetonitrile, maintained at 0 °C, and a 40 µl/min flow rate. MS data were gathered using an MS$^E$ method in resolution mode with an extended range enabled to avoid detector saturation and maintain peak morphologies. MS was initially calibrated with [Glu1]-Fibrinopeptide B human (Glu-fib) peptide, and MS data were gathered with lock mass correction using Glu-fib peptide in positive polarity.

Peptide identification was performed by *ProteinLynx Global Server* (PLGS, 3.0.3, Waters), and identified peptides were filtered by *DynamX* Software (Waters Corp., 176016027.) according to the following parameters: minimum intensity of 800, the minimum sequence length of 3, maximum sequence length of 20, minimum products of 2, minimum products of per amino acid of 0.1, minimum score of 6, maximum MH+ error threshold of 0 ppm. Only high-quality peptides were used for analyses. Deuterium uptake was calculated by an increase in mass/charge ratio compared to $t_0$. Coverage map and heat-map graphs were generated by *DynamX* Software and heat-map data was visualized by mapping the heat-map data on structure via PyMOL. Relative Fractional Uptake graphs were generated by GraphPad Prism 8.1 according to uptake data obtained from *DynamX* Software (Waters Corp., 176016027). Accordingly, the heat maps were generated and the structural features based on deuterium uptake rate were visualized by *PyMOL* (www.schrodinger.com/pymol) [29].

## Gaussian network model in the INSv and IAsp monomers

The Gaussian Network Model (GNM) is a normal mode analysis (NMA)-based approach that provides accurate predictions of protein motions, catching global dynamics in slow modes and local motions in fast modes. In slow modes, residues with high fluctuations represent flexible regions, while residues with low fluctuations define hinge sites that are critical for protein activity [30]. The fast modes highlight "kinetically hot residues," which are highly fluctuating residues constrained within the structure and involved in folding or binding. We conducted normal mode analysis using ProDy [30]. Specifically, we employed the GNM analysis for the newly determined INSv structure at ambient temperature and a previously resolved structure at cryogenic conditions (PDB ID: 4GBN). Contact maps were defined based on all Cα atoms, utilizing a cutoff distance of 8.0 Å. N − 1 nonzero normal modes were derived, where N represents the total number of Cα atoms. For each structure, we calculated squared fluctuations over the weighted 10 slowest and 10 fastest modes, enabling a comparison of their global and local motions, respectively

## Differential scanning calorimetry of monomer INSv and IAsp

Differential scanning calorimetry (DSC) was conducted to assess the thermal stability of IAsp and INSv insulin samples at a 2 mg/ml concentration, employing a SETARAM Micro DSC-III calorimeter (Jandel Scientific). The measurements were carried out within the 20–100°C temperature range, employing a heating rate of 0.3 K/min. Precision balancing, with an accuracy of ± 0.05 mg, was employed for the IAsp and INSv samples and the reference (20 mM citric acid buffer solution) to mitigate corrections involving the heat capacity of the vessels. A secondary thermal scan of the denatured sample was performed for baseline correction. Analysis of the melting temperature ($T_m$), calculation of calorimetric enthalpy ($\Delta H_{cal}$), and determination of the change in entropy ($\Delta S$) were executed using the OriginPro® 2022b software (OriginLab Corporation, Northampton, USA).

## Hexamerization and crystallization of INSv insulin analog

The transformation of the INSv sample from its monomeric state to the $Zn^{2+}$-bound form involved a $Zn^{2+}$ precipitation step using a solution comprising 2.4 M NaCl, 100 mM citric acid, and 6 mM $ZnCl_2$. Subsequently, solubilization was achieved by using 25 mM Tris-HCl solution at pH 8.0. The $Zn^{2+}$-bound INSv samples were then screened for crystallization using sitting drop, microbatch under oil method. Approximately 3000 commercially available sparse matrix and grid screen crystallization conditions were employed for the initial crystallization screening [31]. In a 72-well Terasaki plate (Cat#654,180, Greiner Bio-One, Austria), equal volumes of crystallization conditions (v/v, 0.83 µl) were mixed in a 1:1 ratio with a 5 mg/ml INSv solution. Each well was covered with 16.6 µl of paraffin oil (Cat#ZS.100510.5000, ZAG Kimya, Türkiye) and then incubated at 4°C. Crystal formation in the wells of the Terasaki plates was observed using a compound light microscope.

**Sample delivery and data collection.** Data collection was performed using Rigaku's XtaLAB Synergy Flow X-ray diffraction (XRD) system, controlled by *CrysAlisPro* software (Rigaku Oxford Diffraction, 2022), following the protocol outlined in Atalay et al. (2022) [32]. The airflow temperature of Oxford Cryosystems's Cryostream 800 Plus was set to 300 K (26.85 °C) and maintained consistently at ambient temperature during data collection. The 72-well Terasaki plate was mounted on the adapted XtalCheck-S plate reader attachment on the goniometer omega stage [33]. Two dozen crystals were used for the initial screening to assess diffraction quality. Adjustments to omega and theta angles and X, Y, and Z coordinates were made to center crystals at the eucentric height of the X-ray focusing region. Subsequently,

diffraction data were collected from multiple crystals. Crystals demonstrating the highest resolution Bragg diffraction peaks were chosen for further data collection, and exposure times were optimized accordingly to minimize radiation damage during overnight data collection. The most promising crystals were cultivated in a buffer solution containing 2.4 M NaCl, 100 mM Tris-HCl at pH 7.4, 6 mM $ZnCI_2$, 20 (w/v) poly (ethylene glycol) PEG-8000. Throughout the data collection process, *XtalCheck-S* was configured to oscillate to the maximum extent permitted by the detector distance, aiming to optimize crystal exposure oscillation angles. 42 frames were captured, each running lasting 3 minutes (5 seconds per frame) for all individual crystals. Parameters were set with the detector distance at 100.00 mm, a scan width of 0.5 degrees oscillation, and an exposure time of 10.00 seconds per image.

**Data processing and structure determination.** Following the optimization of plate screening parameters for all crystals, a 42-degree data collection was conducted for each selected crystal. All crystals were organized in *CrysAlisPro* for comprehensive data collection. An optimal unit cell was selected, and peak identification and masking procedures were executed on the collected data. Subsequently, a batch script incorporating the `xx proffitbatch` command was generated for cumulative data collection. The batch data reduction was accomplished using the *CrysAlisPro* suite through the script command, producing files containing all integrated, unmerged, and unscaled data (*.rrpprof) for each dataset. To merge all datasets into a reflection data (.mtz) file, the *proffit merge* process in the *Data Reduction* section of the main window of *CrysAlisPro* was employed. The reduced datasets (.rrpprof files) were further merged using *proffit merge*. The entirety of the data underwent refinement, integration, and scaling processes through the aimless and pointless implementations within *CCP4* [34]. Ultimately, the processed data were exported into *.mtz formats. The crystal structure of INSv was determined under ambient temperature within the R3:H space group, employing the automated molecular replacement tool *PHASER* integrated into the *PHENIX* software package [35]. Utilizing a previously published X-ray structure (PDB ID: 4GBN) [36] as the initial search model, the coordinates from 4GBN were employed for the preliminary rigid-body refinement within *PHENIX*. Subsequent refinement steps included simulated-annealing refinement, adjustment of individual coordinates, and Translation/Libration/Screw (TLS) parameters. Furthermore, a refinement technique known as composite omit map, embedded in *PHENIX*, was executed to identify potential locations of altered side chains and water molecules. The final model underwent meticulous examination and model-building in *COOT*, focusing on positions exhibiting notable difference density. Extraneous water molecules located beyond regions of significant electron density were manually excluded. The all crystal structure figures were visualized using *PyMOL* ([www.schrodinger.com/pymol](www.schrodinger.com/pymol)) and *COOT* [37].

## Statistics

Deuterium uptake data report was obtained from DynamX Software after HDX-MS analysis and obtained time-dependent Relative Fractional Uptake (RFU) values were presented as XY graphs generated by GraphPad Prism 8.1. The results of calcium imaging were further processed and analyzed using Excel (Microsoft) and Prism 9 (GraphPad Software, La Jolla, CA, USA), with quantification performed using the parameters of Area Under the Curve (AUC) and the number of active cells. The total AUC was calculated as the total area beneath the time-series curve. Active cells were manually enumerated. Temporal variations in light intensity within the selected cell bodies were graphically represented over time. Pairwise comparisons of calcium imaging data between the IAsp and INsv groups were performed using Student's t-test, with p-values below 0.05 considered statistically significant.

## Results and discussion

To enhance the flexibility of insulin, we introduced double substitution from arginine (Arg) to lysine (Lys) and proline (Pro) to aspartic acid (Asp) in chain B. The internal Arg43 (ArgB22) on insulin was specifically mutated to Lys to mitigate the formation of undesired cleavage products, as trypsin exhibits a higher affinity for Arg than Lys. INSv was produced (S1A Fig) and subsequently folded through a refolding process in E. coli Rosetta™ 2 Competent Cells, which contained a double mutant INSv vector. The circular dichroism (CD) experiment primarily monitored changes in ellipticity in the far-UV region (195–250 nm) to elucidate the secondary structure of recombinant INSv (S1B Fig). Two local minima at 208 nm and 222 nm, predominantly observed at pH 9.5 and pH 10.0, indicate a substantial presence of alpha-helix secondary structure. At low pH 3.0, the circular dichroism results show an alteration in the alpha-helical secondary structure.

### Novel double-substituted INSv induces cellular activity

Calcium imaging was performed to verify whether the INSv monomer induces any cellular activity compared to the IAsp monomer (Fig 1).

Calcium is an essential intracellular messenger in neurons; consequently, while most neurons exhibit a resting intracellular calcium concentration ranging from approximately 50–100 nM, this concentration may transiently escalate by a factor of 10–100 during electrical activity [38]. Intracellular calcium signals operate a broad spectrum of processes, from neurotransmitter release on a microsecond scale to gene transcription occurring over minutes to hours. In examining spontaneous activity, we measure the number of events, which are the occurrences noted at each Region of Interest (ROI). Additional parameter we analyze is the Area Under Curve (AUC), highlighted in red on the plot of spontaneous activity (Figs 1A and S2).

Upon the individual administration of INSv vs. IAsp monomers to the cells, the count of cells demonstrating calcium activity increased (Figs 1B and S3 and Movies S1 and Movie S2), accompanied by a decrease observed in the AUC (Figs 1A and S2).

The INSv and IAsp monomers initiated the neural networking, resulting in a decrease in AUC (Figs 1A and S2), which is expected as the event digit decreases. However, while the AUC decrease in IAsp is deemed insignificant, that observed in INSv is considered significant (S2 Fig).

### Alternate structural status in IAsp and INSv hexamers

We determined the ambient temperature structure of INSv at 2.5 Å resolution (Table 1 Figs 2A-C and S4).

For comparison, the IAsp structure, determined under cryogenic conditions at pH 6.5, was used as the reference model, whereas INSv crystals were grown at pH 7.4. Two nearly identical monomer molecules within the asymmetric unit cell exhibit an overall RMSD of 0.81 Å. Superpositioning and comparative analysis with the PDB structure of 4GBN revealed overall RMSD values ranging from 0.3 Å to 0.83 Å for the nearly identical monomer molecules in the asymmetric unit. Molecular replacement with a dimer in the asymmetric unit facilitated the solution of the crystal structures of INSv insulin at pH 7.5. Specifically, amino acid residues 1–8 exhibited a "T" conformation in one monomer and an "R" conformation in the other with well-defined electron densities. In the previously reported IAsp structure (PDB_ID: 4GBN) (Fig 2B-D), three symmetry-related "TR" dimers constituted the biological unit [36]. The hexameric configuration of INSv is organized near a crystallographic 3-fold axis aligned with the central channel [36]. In contrast to the 4GBN structure (Fig 2B), a distinctive feature of

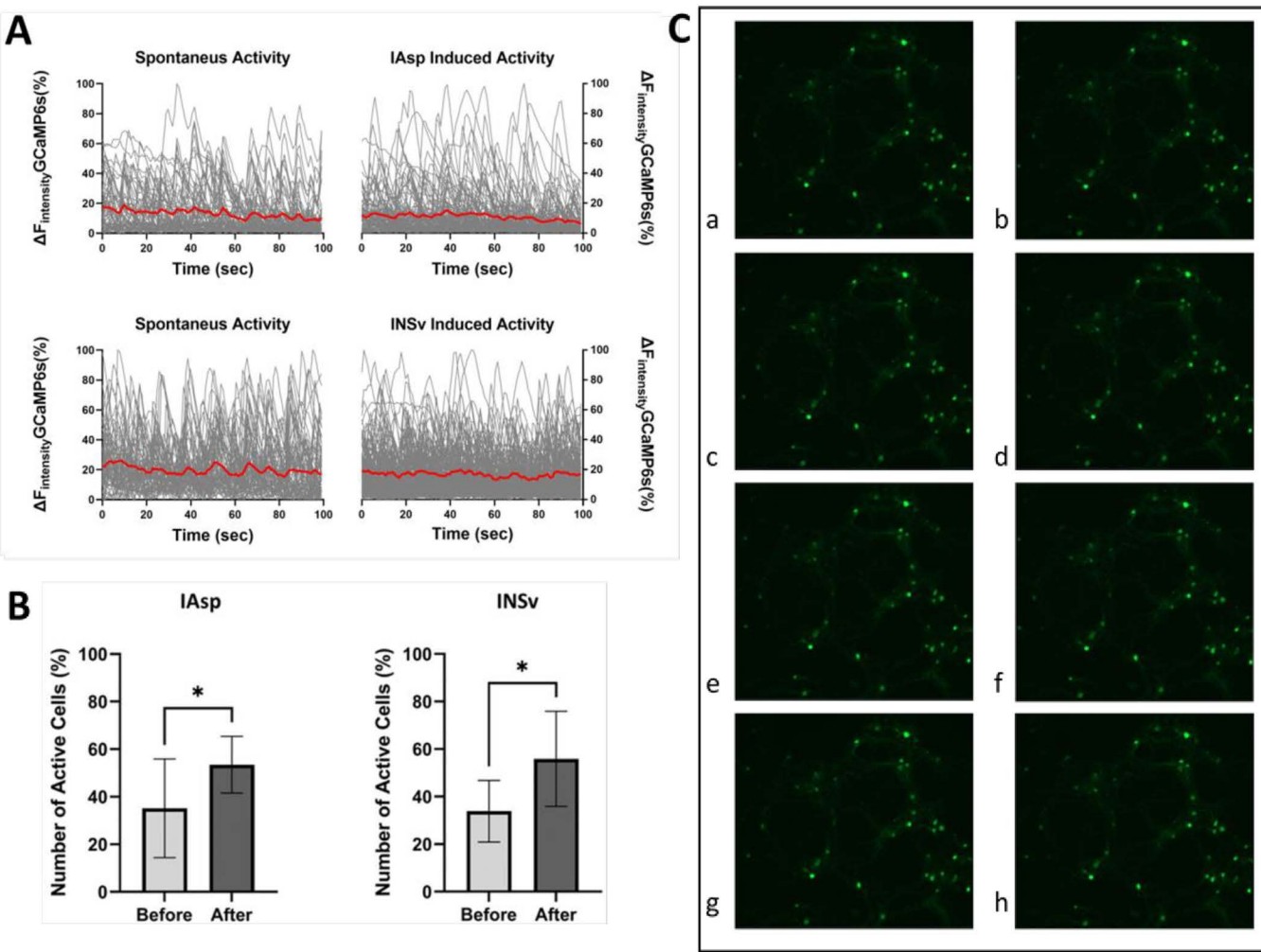

**Fig 1. Comparative analysis of calcium imaging between INSv and IAsp.** (A) The graph displays temporal light intensity shifts indicative of recorded calcium activity. The under areas of the red trend have highlighted the area under the curve (AUC), which indicates the average of the areas under the light intensity change curve recorded from each ROI. Statistical analysis of the AUC measurement has been indicated at S2 Fig (B) Statistical analysis of the effects of INSv and IAsp on the proportion of active neural cells was performed using an unpaired t-test, yielding a p-value of 0.0358 for INSv and 0.0473 for IAsp. (C) Sequential visualization of calcium activity propagation following INSv stimulation includes **a in C panel,** a pre-stimulation network image. **b-h in C panel,** a progressive spread of calcium activity across the network, with detailed annotations for each stage in S3.Fig.

the INSv hexamer involves the outward orientation of Asn3 (Fig 2A) in chain B relative to the interior of the channel. This spatial arrangement suggests less stable electrostatic interaction with the two symmetry-related B3Asn residues (Fig 2A). This outward orientation of Asn3 of INSv compared to IAsp may also be due to the temperature differences during the data collection, but thermal displacement (b-factor) analysis and computational GNM analyses of the INSv and IAsp crystal structures closely resemble the dynamics observed in *in-solution* analyses of monomer INSv and IAsp (such as HDX-MS, DSC). The crystal structures of INSv and 4GBN exhibit closely similarities in both Cα atoms and side chains. Although there are positional variations, these are primarily minor, with relatively major discrepancies observed in the side chains of 4GBN [36] IAsp compared to INSv. Upon superimposing the current INSv insulin structure onto 4GBN in the $T_3R_3$ conformation, a dissimilarity becomes evident in the region encompassing amino acid residues 1–8 in chain B. This dissimilarity is observed in both backbone and side chain conformations, providing evidence for INSv insulin adopting

**Table 1. Data collection and refinement statistics of *INSv* insulin analog.**

| Data collection | *INSv* |
|---|---|
| Instrument | *Turkish DeLight* |
| Resolution range | 21.64–2.501 (2.59–2.501) |
| Space group | R 3: H |
| Unit cell | 80.376 80.376 38.04 90 90 120 |
| Redundancy | 24.9 (12.9) |
| Completeness (%) | 97.70 (91.02) |
| Mean I/sigma(I) | 6.88 (1.72) |
| CC1/2 | 0.952 (0.358) |
| CC* | 0.988 (0.726) |
| R-work | 0.3069 (0.3590) |
| R-free | 0.3529 (0.4477) |
| Protein residues | 102 |
| RMS (bonds) | 0.091 |
| RMS (angles) | 0.86 |
| Ramachandran favored (%) | 94.68 |
| Ramachandran allowed (%) | 4.26 |
| Ramachandran outliers (%) | 1.06 |
| Average b-factor | 55.83 |
| Macromolecules | 56.44 |
| Ligands | 38.47 |
| Solvent | 45.06 |

the $T_3R_3$ conformation with a pure TR-state dimer configuration. A pairwise global structural alignment between the $T_3R_3$-INSv and -4GBN IAsp structures highlights their similarity, with a global RMSD slightly over 0.8 Å. This similarity persists, indicating a consistent outcome attributable to the $Zn^{2+}$-bound form.

We conducted a separate b-factor analysis from GNM for the monomers of 4GBN and INSv (Fig 3).

In the R-forms of the INSv and 4GBN IAsp structures (Fig 3A-C), it is observed that the N-terminal segments ranging from AsnA21-Gln25(GlnB4) and Gly41(GlyB20)-Glu42(GluB21) of INSv's B chain exhibit slightly higher fluctuations compared to their counterparts in the 4GBN IAsp structure. Similarly, in the T-forms of the INSv and 4GBN IAsp structures (Fig 3B-D), the regions spanning Cys40(CysB19)-Glu42(GluB21) in INSv demonstrate pronounced flexibility relative to the T-form of the 4GBN IAsp structure. On the other hand, the R-form of INSv (Fig 3C) is greatly compatible with our H/D reactions pattern we observed in Fig 4A, assuming INSv monomer in-solution is probably a more flexible R-form, thus implying that the R-forms of INSv monomer (Fig 3C) exhibit greater fluctuation than its T-form (Fig 3D). Additionally, substituting R with K might induce a flexible R-state despite the limited influence of $Zn^{2+}$ coordination on the hexameric form.

## HDX kinetics of INSv is compatible with crystal structure dynamics

In this study, we quantified deuterium exchange in specific regions of INSv and IAsp, identified the differences in locations, and analyzed variations in their local dynamics. For HDX-MS experiments, we used the monomeric form of insulin to compare monomer kinetics under identical experimental conditions (S1 Table).

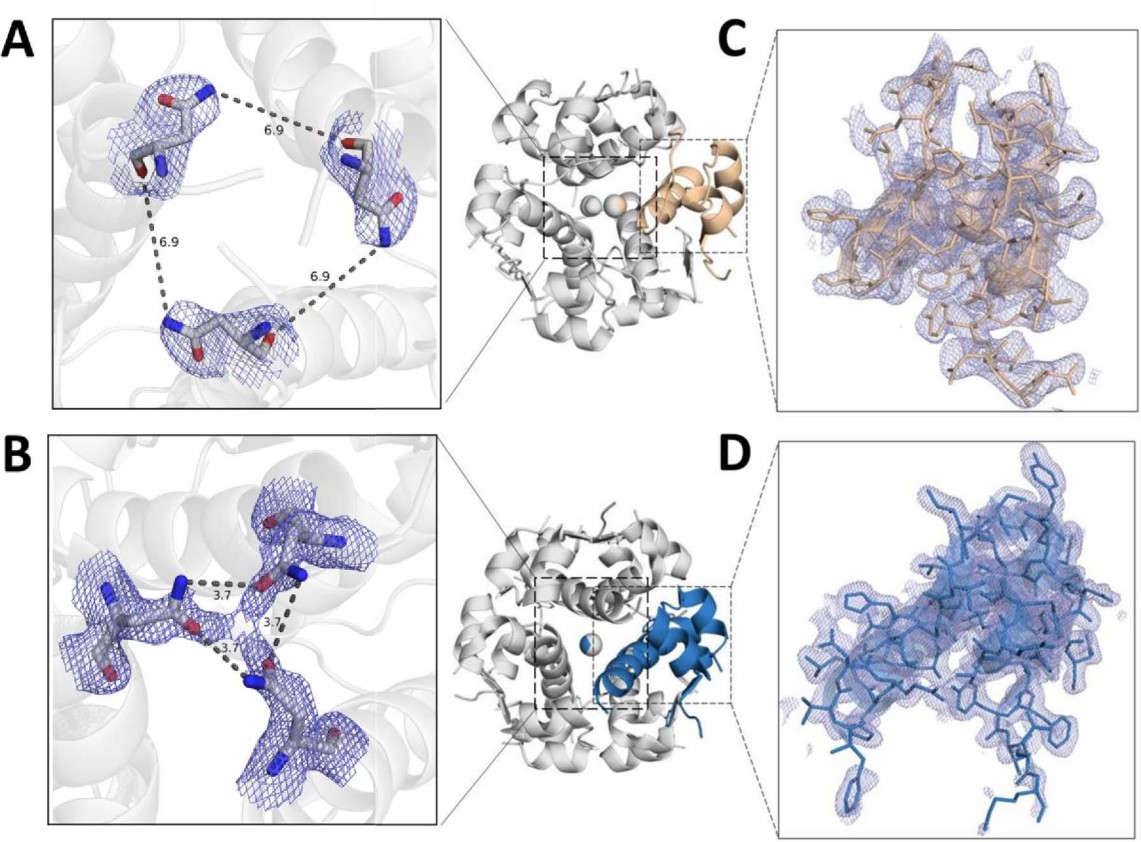

**Fig 2. A comparison between the INSv and 4GBN IAsp structures, presenting both distant and close-up views.** Panels (**A**) and (**B**) depict the electrostatic interaction distances of AsnB3 residues within each monomer of insulin hexamers, INSv and 4GBN, respectively. Additionally, panels (**C**) and (**D**) examine a monomer from insulin hexamers INSv and 4GBN, respectively. 2*Fo-Fc* simulated annealing-omit map at 1 sigma level is colored in slate.

The insulin monomer consists of two distinct chains: chain A (21 residues) and chain B (30 residues). Additionally, GNM analysis (using the PDB file of INSv and IAsp crystal structures) was also conducted to observe if there is a similarity between in-solution monomer kinetics and in-crystallo monomer dynamics. Consequently, even if substantial kinetic differences in H/D exchange reaction were not observed in the mutated region of these insulins, notable differences were noted in the kinetics of the remaining monomeric regions (Figs 4A and 5A).

Initially, to ensure uniformity in pH conditions during the exchange reaction, Tris-HCl buffer at pH 9.1 (pD 9.5) was selected, owing to its ability to maintain insulin's *in vivo* monomeric state [11,12]. The temporal evolution of deuterium exchange profiles by IAsp is indicated in Fig 4. In the H/D exchange reaction of the IAsp monomer, certain residues, namely regions involving CysA11-LeuA13, AsnA21-Cys28 (CysB7), Glu34(GluB13)-Tyr37(TyrB16), Arg43(ArgB22)-Gly44(GlyB23), and Asp49(Asp28), exhibit a notable degree of deuterium exchange indicating relatively more mobile compared to other residues (Fig 4A), consistent with findings from computational analyses (Fig 4B). Our normal mode analysis identifies IleA2, CysA6, IleA10, GlnA15-LeuA16, AsnA18-TyrA19, Leu27(LeuB6) to Cys28(CysB7), Glu34(GluB13) to Leu36(LeuB15), Leu38(LeuB17) to Val39(ValB18), and Arg43(ArgB22) to Gly44(GlyB23) as regions undergoing dynamic dissections (Fig 4B, left), and these are

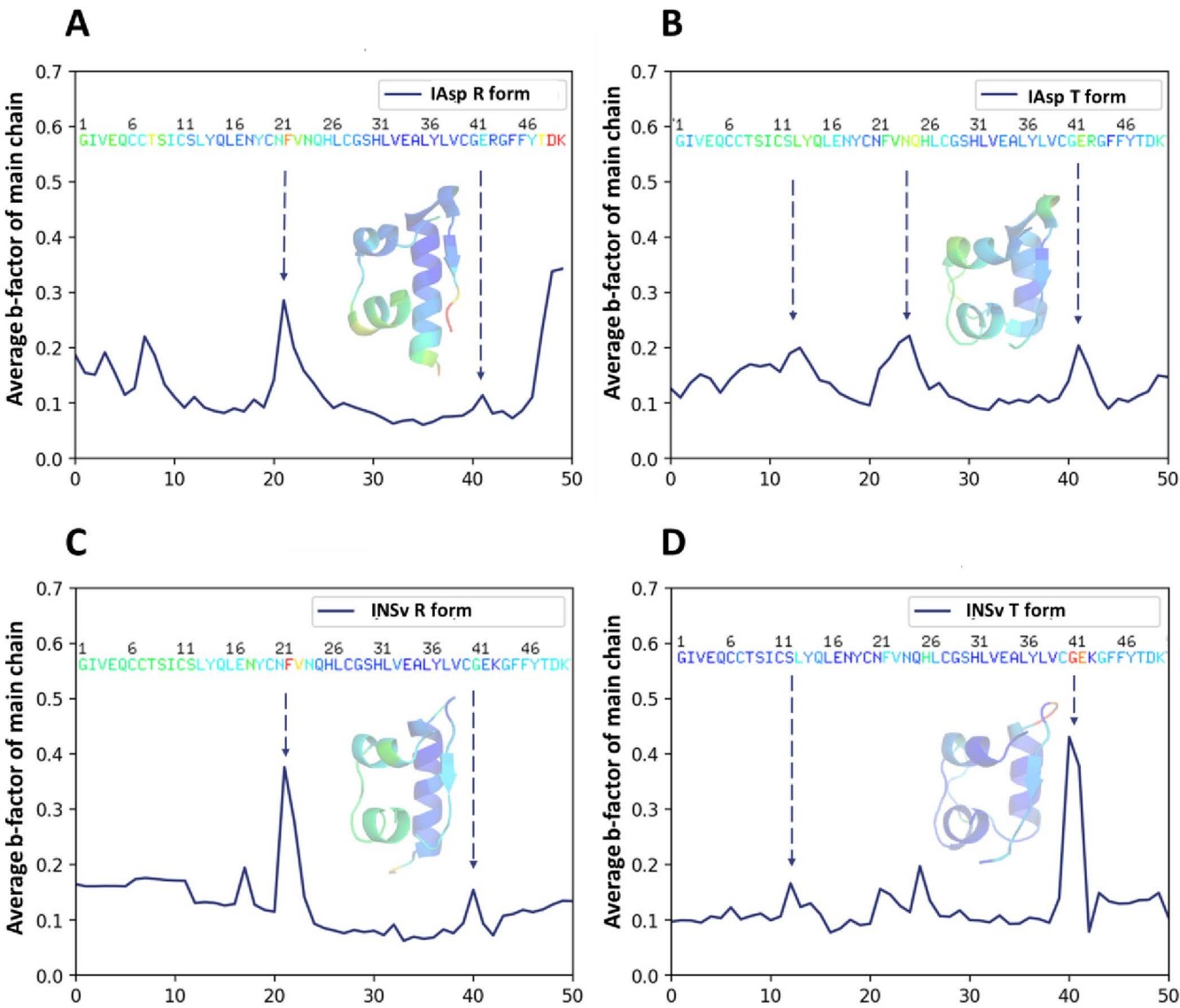

**Fig 3. Experimental b-factor analysis and cartoon representation of each 4GB IAsp and INSv dimers' monomer.** (**A,** B) Thermal distance analysis supports the experimental b-factor of the IAsp structure's R-state and T-state conformers. (**C, D**) Thermal distance analysis supports the experimental b-factor of the INSv structure's R state and T state conformer. Cartoon mode and amino acid letters are colored by the "b-factor" option, which is implemented in PyMOL. The blue color indicates areas of rigidity, whereas the red color indicates regions of flexibility. *Insulin monomer is referred to as a single chain throughout the b-factor from X-ray and GNM analyses.*

referred to as kinetically hot regions (Fig 4B, right). These identified hinge sites from our computational analysis are compatible with observations in the HDX-MS heat-map results. However, SerA12 and GlnA15-LeuA16 residues are identified as crucial hinge regions in only GNM analysis (Fig 4B).

Next, the H/D exchange reaction in INSv insulin analog was compared to IAsp to assess potential distinctions (Fig 5). The INSv analog demonstrated higher deuterium exchange compared to IAsp (after a 1440-minute reaction, n = 3), displaying a more kinetic conformation for INSv analog compared to IAsp, agreeing with the findings of B-factor of INSv crystal structure (Fig 3C) and DSC analysis (Fig 6A).

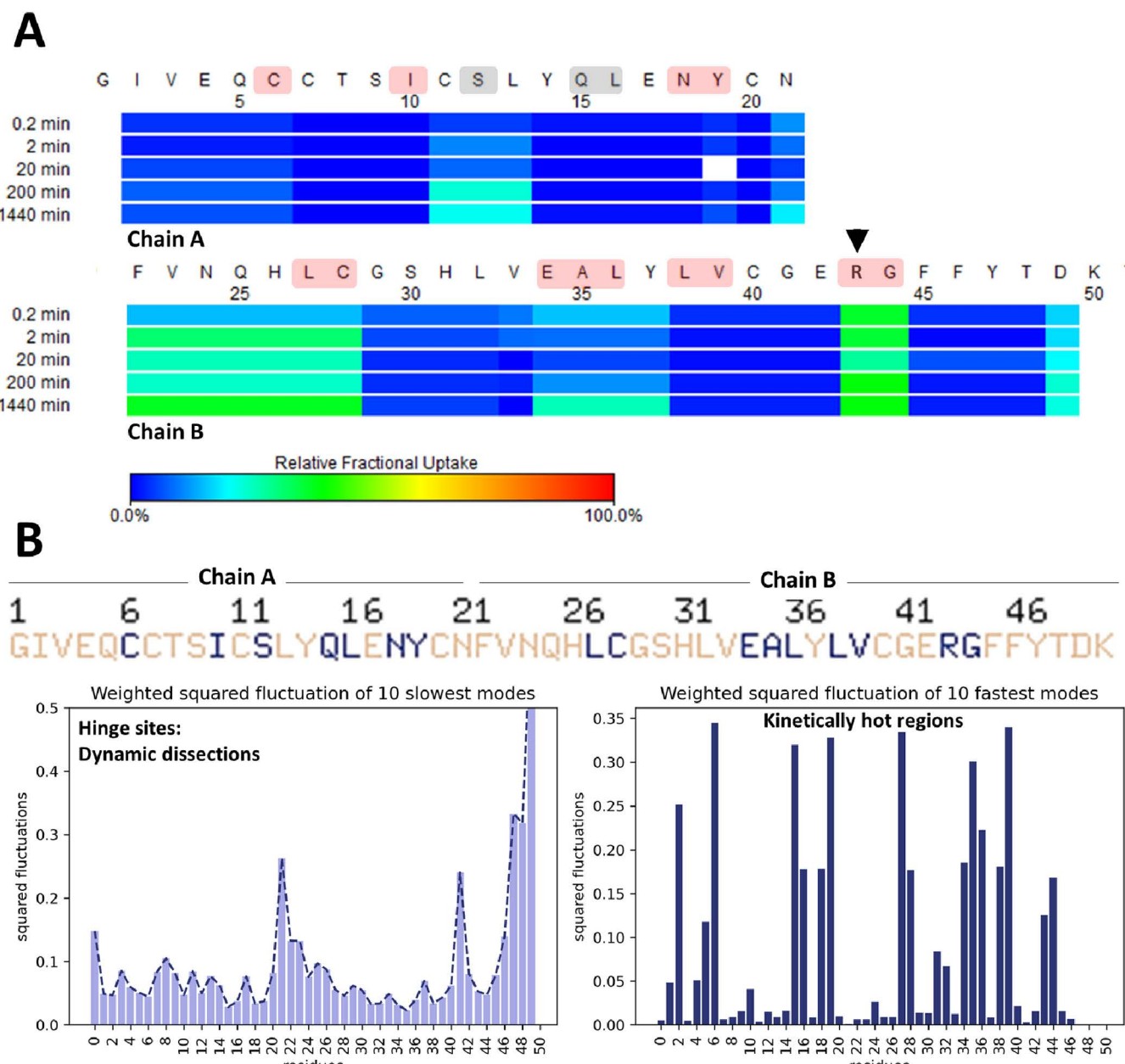

**Fig 4. Complementary experimental and computational dynamic analysis of IAsp.** (A) Heat map displaying time-course H/D exchange measurements of IAsp. The color code from blue to red describes deuterium exchange compared to $t_0$. The black arrow indicates the original residue that is not mutated (B) Cumulative 10 slowest and 10 fastest modes from GNM to show dynamic dissections and kinetically hot regions. The critical residues for relative fractional deuterium exchange, which are highlighted in red (in panel A), are largely consistent with the cumulative 10 slowest and 10 fastest modes from GNM, which are indicated by blue letters (in panel B). The critical residues observed in the GNM alone, not in H/D reaction, are illuminated by gray color in panel A. *Insulin monomer is referred to as a single chain throughout the HDX-MS and GNM analyses.*

Within the H/D exchange reaction of the INSv monomer, specific regions, IleA2-CysA6, SerA9-AsnA18, CysA20-AsnA21, and Val39 (ValB18)-Asp49 (AspB28), exhibited a noticeable degree of relative mobility in contrast to other residues (Fig 5A), broadly consistent with computational analyses (Fig 5B). Our normal mode analysis identified GluA4-CysA6, SerA9-IleA10, AsnA18-TyrA19, His31 (HisB10)-Leu32 (LeuB11), and Val39 (ValB18) residues as

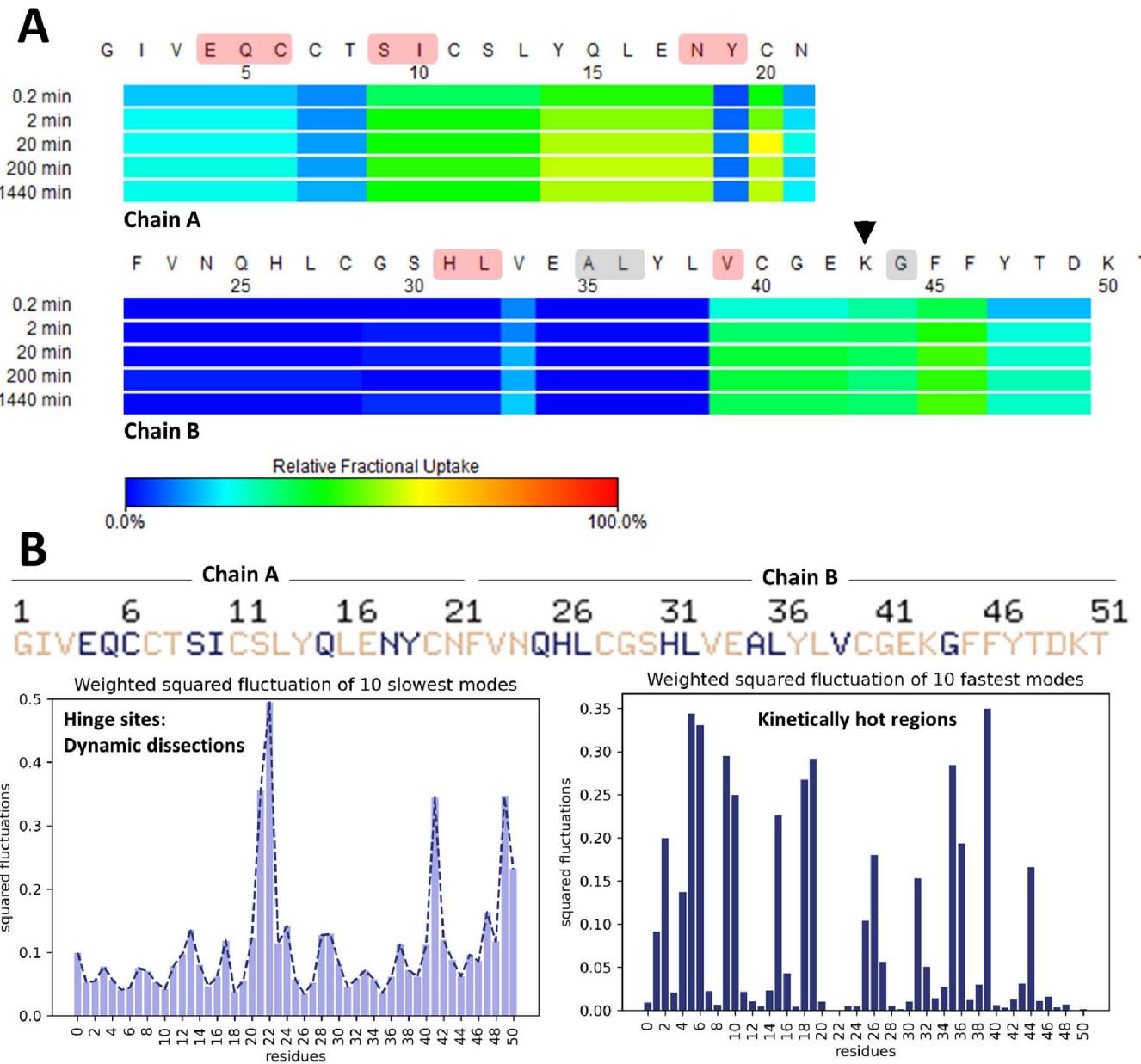

**Fig 5. Complementary experimental and computational dynamic analysis of INSv.** (A) Heat map displaying time-course H/D exchange measurements of INSv. The color code from blue to red describes deuterium exchange compared to $t_0$. The black arrow indicates the mutated residue (R→K) (B) 6Cumulative 10 slowest and 10 fastest modes from GNM to show dynamic dissections and kinetically hot regions. The critical residues for relative fractional deuterium exchange, which are highlighted in red (in panel A), are largely consistent with the cumulative 10 slowest and 10 fastest modes from GNM, which are indicated by blue letters (in panel B). The residues observed in the GNM alone, not in the H/D reaction, are illuminated in gray color in panel A. *Insulin monomer is referred to as a single chain throughout the HDX-MS and GNM analyses.*

regions undergoing dynamic dissections (Fig 5B, left), denoted as kinetically hot regions (Fig 5B, right). Notably, the identified hinge sites from our computational analysis align seamlessly with observations in the HDX-MS heat-map results. Nevertheless, Ala35 (AlaB14)-Leu36 (LeuB15) and Gly44 (GlyB23) residues observed as crucial hinge regions only in our GNM analysis (Fig 5B).

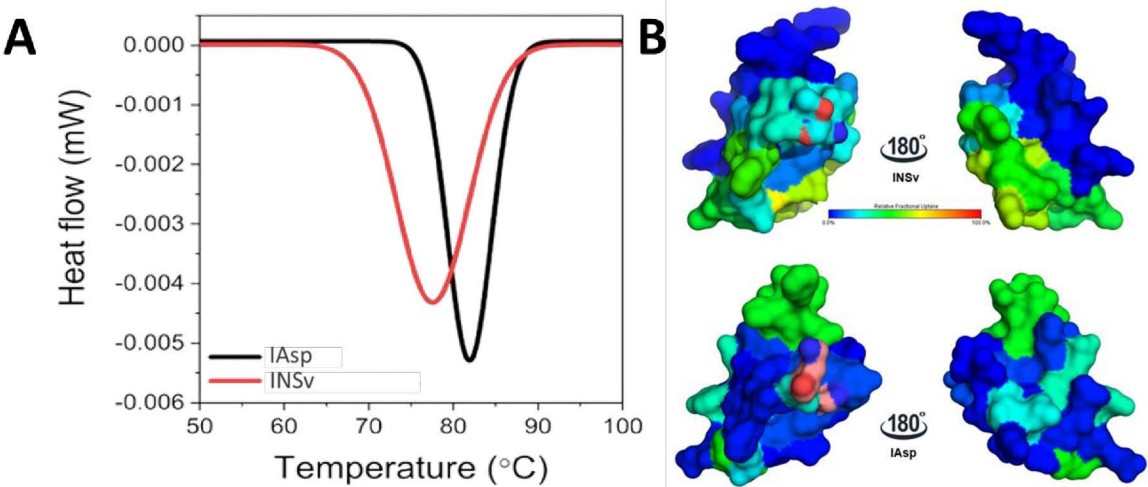

**Fig 6. Representation of thermal stability analysis comparing to H/D dynamics variability of IAsp and INSv analogs.** (A) The variation in heat flow depends on the melting temperature ($T_m$), which signifies the point at which 50% of the protein undergoes denaturation. Furthermore, the calorimetric enthalpy ($\Delta H_{cal}$) was computed, and the alteration in conformational entropy ($\Delta S$) was ascertained by evaluating the ratio of $\Delta H_{cal}$ to $T_m$ in Kelvin (K). (B) Visualization of deuterium exchange over 1440-minute intervals through aligning our newly determined monomer of INSv with the monomer of IAsp. The blue color marks the areas of rigidity, whereas the red color indicates regions of flexibility.

S5A Fig depicts the close-up view of deuteration levels of the corresponding fragments of INSv and IAsp over the 200-minute time course.

The uptake profiles of most fragments exhibited variations between INSv and IAsp (S6 Fig, S7 and S8).

Additionally, there was a minimal resemblance in the fragment encompassing His31(HisB10) to Asp49(AspB28), which includes the mutation site in INSv (S5B Fig).

### Thermal stability analysis confirms distinctive H/D kinetics of INSv

Differential scanning calorimetry (DSC) measurements were conducted to investigate the thermal stability of the INSv compared to commercially available IAsp. DSC measurements revealed significant differences in the thermodynamic properties of IAsp and INSv (Fig 6A). The thermal unfolding of IAsp shows a steep endothermic denaturation with a $T_m$ of 81.9 °C, a $\Delta H_{cal}$ of 35.46 KJ/mol, and a $\Delta S$ of 0.099 KJ/mol·K. On the other hand, the $T_m$ of the INSv was considerably lower ($T_m$ = 77.6 °C), and the $\Delta H_{cal}$ also decreased ($\Delta H_{cal}$= 31.55 KJ/mol), as well as the $\Delta S$ showed increasing tendency ($\Delta S$= 0.142 KJ/mol·K). The significantly lower $T_m$ suggests a less stable and most likely, less compact conformation for the INSv. Moreover, the decreased $\Delta H_{cal}$ and increased $\Delta S$ of INSv align with the lower $T_m$, assuming the protein conformation is more flexible (Fig 5A). This DSC interpretation is also supported by the distinct H/D exchange kinetics observed between IAsp and INSv (Figs 4A and 4B), suggesting that INSv exhibits comparable flexibility to IAsp in-solution (Fig 6B).

The $Zn^{2+}$-coordinated hexameric form of insulins exemplifies the characteristic structural model inherent in globular proteins [39]. Serving as a pivotal model in the evolution and application of X-ray crystallographic techniques [8], the hexamer functions as a stable reservoir of the hormone within both the secretory granules of pancreatic β-cells and pharmaceutical formulations [40]. The integration of structure-based mutagenesis into the $Zn^{2+}$-bound insulin hexamer [41] has significantly advanced the development of rapid-acting insulin analog formulations suitable for prandial injection and pump use [42,43]. Despite their initial

categorization as "monomeric" [36], these analogs are presently designed either as $Zn^{2+}$-bound insulin hexamers or as an equilibrium distribution of $Zn^{2+}$-free oligomers [2]. Although it assembles into dimers and hexamers during its biosynthesis and storage, insulin is always active as a monomer [1]. Even though X-ray analysis can reveal the 3-dimensional structure of the insulin monomer states [44,45], studies about the monomer structure and dynamics in solution are severely limited by insulin self-association into dimers and higher oligomers.

The present study has explored the examination of the functional (Figs 1, S2 and S3) INSv alongside the IAsp monomers to illuminate the fundamental structure and dynamics of these rapid-acting insulin in a solution. The substitution of Arg with Lys at position B22 and Pro with Asp at position B28 distinguish the INSv, which DSC confirmed to be a less stable insulin analog compared to IAsp (Fig 6A). The observed reduction in $\Delta H_{cal}$ for INSv, combined with an increased $\Delta S$ (entropy), implies greater conformational mobility or a broader ensemble of accessible states. Lys and Arg, both positively charged basic amino acids under physiological conditions, are predominantly exposed on the surfaces of proteins [46–48]. Despite their shared function as basic residues, Arg contributes more stability to protein structure than Lys, attributed to its distinctive geometric structure [47]. In contrast, the basic functional group of Lys allows interaction in only one direction [49–51]. This capability enables Arg to form more electrostatic interactions, including salt bridges and hydrogen bonds, than Lys [52]. Presumably, this results in stronger interactions than those generated by Lys.

ArgB22 was supposed to be pivotal for insulin function due to its effective genetic invariance [52–54]. The anomalously low activity of guinea pig insulin in various standard bioassays had been attributed to substituting aspartic acid for ArgB22 position [55]. A study revealed that the altered orientation and flexibility of the B20-B23 β-turn can interfere with forming disulfide bonds within proinsulin carrying the GlnB22 mutation. Assumed that this mutation can negatively impact the WT proinsulin that is simultaneously biosynthesized in β-cells, and this interference might play a significant role in the development of Maturity-Onset Diabetes of the Young in patients producing [GlnB22]-insulin [56]. Nonetheless, the studies are controversial. In a study describing the modification of the insulin ArgB22 residue by an N8N9-(1,2-hydroxycyclohex-1,2-en) group and an adipoyl group, observed that the modified insulin displayed specific activity indistinguishable from WT insulin when tested for enabling lower blood glucose concentration [57]. The preservation of biological activity challenges the previously widely accepted notion that the ArgB22 residue is essential for the hormone's activity. A similar study produced a recombinant monomeric B22Asp desB30 insulin analog, aiming to explore a more cost-effective alternative for producing monomeric human insulin analogs to treat diabetes. These analogs have demonstrated a propensity for the monomeric form and exhibit 30–40% of the activity compared to the endogenous human insulin [58]. Moreover, a human insulin analog was synthesized, characterized by replacing the arginine to lysine in B chain. This [Lys-22B] insulin, produced with high efficiency, demonstrated a potency of 13–14 IU mg-1 when assessed using the mouse convulsion method, in contrast to the 23–25 IU mg-1 potency observed for the native hormone [59]. This suggests that, despite the INSv analog demonstrating comparable or even superior activity to IAsp (Figs 1 and S2), its affinity for the insulin receptor is relatively diminished, likely attributable to reduced structural stability.

Based on the compatibility between the experimental b-factor from X-ray analysis (Fig 3C-D) and the H/D exchange reaction of INSv (Fig 5A), the solution structure of this variant closely mirrors the in-crystallo R state (Fig 3C). However, unlike the IAsp structure (Fig 2B), a distinct feature of the INSv hexamer in the R form involves the outward orientation of Asn3 in chain B relative to the interior of the channel (Fig 2A). This spatial configuration leads to a less stable electrostatic interaction with the two symmetry-related AsnB3 residues.

Additionally, the unique R-state of the INSv crystal structure is evident in a flexible INSv monomer according to the b-factor analysis (Fig 3C), which correlates with the H/D exchange reactions for the INSv monomer (Fig 5A). Characterizing INSv and IAsp as native-like monomers at pH 9.0 [11,12,60] and without an organic co-solvent provides an opportunity for a quantitative assessment of amide-proton exchange in $D_2O$. HDX-MS data for INSv indicated swift deuterium exchange at the initial time points in the majority of peptides, followed by local exchange after specific time points (Fig 5A). In contrast, IAsp exhibited no deuterium exchange in the first 35 residues, with only the Phe22(PheB1)-Cys28(CysB7) and Arg43(ArgB22)-Gly44(GlyB23) peptides showing deuterium exchange and local exchange at 2-min, 20-min, 200-min and 1440-min points (Fig 4A). Kinetic variability between two analogs in-solution (it means HDX-MS analysis) (Figs 4A and 5A) aligns well with that in-crystallo (it means XRD) (Figs 2 and 3A-C). According to b-factor analysis from X-ray crystallography, significant mobility is observed up to the 26th residue in the R state form of INSv (Fig 3C), which is clearly reflected in the H/D exchange map (Fig 5A). Similarly, the higher mobility near the mutation site and the C-terminal region (Fig 3C) is notable and consistent with the H/D exchange data (Fig 5A). As observed in DSC, the lower thermal stability of INSv (Fig 6A) suggests that the INSv seen in the X-ray structure can be characterized by a more dynamic R state (Fig 3C). While DSC measures the global thermodynamic properties of the folded protein, H/D exchange kinetics in-solution and GNM dynamics obtained from in-crystallo provide insight into local and global mobility.

The formation of insulin oligomers primarily relies on establishing noncovalent bonds between interacting residues from adjacent monomers [61]. However, alterations in pH can interrupt these interactions, causing residues in distant sites to cease their communication paths and transition into monomeric states [11,12]. Modifying monomeric states results in substantial modifications to the customary residue communication network compared to oligomeric states [62–64]. These alterations can also be elucidated through computational GNM analysis [30], which is analogous to our insights from experimental H/D exchange reactions (Figs 4B and 5B). We conducted GNM in this study to further elucidate the computational dynamics exhibited by the INSv monomer (Fig 5B) compared to the monomeric form of IAsp (Fig 4B), aiming to correlate HDX-MS results (Figs 4A and 5A). GNM-based methodologies precisely assess global protein motions at lower frequencies and local motions at higher frequencies [63,64]. In the INSv monomer, pivotal hinge sites crucial for insulin activity are identified in residues CysA6, SerA9, AsnA18-TyrA19, and Leu32(LeuB11), as observed through computational GNM dynamics (Fig 5B) and experimental H/D exchange kinetics (Fig 5A). Conversely, in the IAsp monomer, hinge sites critical for insulin activity are in CysA6, IleA10, AsnA18-TyrA19, Cys28(CysB7), Glu34(GluB13), Leu38 (LeuB17), and Arg43(ArgB22)-Gly44(GlyB23) (Fig 4B). Remarkably, these residues (in Figs 4B and 5B) exhibiting dynamic changes are the regions involved in receptor interactions [65].

Finally, observed that both INSv (before, 34% activity; after, 56% activity) and IAsp (before, 35% activity; after, 53% activity) significantly increased the activity of cells (Fig 1 and S2–S3), but they do not differ considerably from each other in the calcium imaging experiment. In both cases, there is an increase in the active cell count due to spontaneous events. Based on a study that used Fura-2 instead of GcaMP6, it was claimed that Fura-2 might not be sufficient to detect all events as well as suggested that the insulin concentration should be increased [66]. Accordingly, our experiment was designed with a relatively higher insulin dosage of 3.4 µM and utilized GcaMP6 instead of Fura-2. Although GcaMP6 and Fura-2 are highly similar, GcaMP6 is more effective at detecting smaller events [67]. Quantitative analysis shows that the pattern of spontaneous activity in INSv is denser than in IAsp, likely due to distinct calcium transients. This indicates that the calcium influx, indicative of the electrical

activity of the cells, is more significant in INSv than in IAsp. Additionally, both analogs acutely induce neural networking, accompanied by a decline in AUC values (Fig 1A-B). The reduction in AUC for IAsp is insignificant (*p*-value: 0.1290), while the decrease in AUC for INSv is statistically significant (*p*-value: 0.0002) (S2 Fig). This indicates that the observed disparity between INSv and IAsp is due to the higher number of events in INSv compared to IAsp (Figs 1B and S2).

All suggest that the distinct underlying mechanisms may be related to the different kinetics and dynamics of INSv and IAsp, as observed *in-vitro* (Fig 1), *in crystallo* (Figs 2 and 3) and *in solution* (Figs 4–6) experiments. The dynamic variations observed, both experimentally and computationally, suggest that while the hexameric structure bound to the $Zn^{2+}$ cation may not change significantly, the substitution of Arg with Lys in a monomer induces distinct kinetic models dependent on particular mobilities. Given the tendency of conformational fluctuations to accelerate the degradation of pharmaceutical formulations, we propose leveraging a concept called "dynamic restructuring" for insulin. This approach could facilitate the strategic development of smart rapid-acting formulations, based on the observed kinetic differences.

## Conclusion

In this study, we introduced a less stable & functional monomeric insulin variant, INSv. The crystal structure of its hexameric insulin variant was determined at a resolution of 2.5 Å at ambient temperature. A H/D exchange dynamic between the IAsp and the INSv revealed that INSv exchanges deuterium atoms more rapidly and extensively than intact IAsp, which is consistent with the diminished hexameric stability of INSv. GNM analysis further supported these kinetic differences. The observed variations prompted exploring the specific regions responsible for both molecules' discrepant H/D exchange reactivity. These deuteration discrepancies, coupled with computational analysis, imply that the R→K substitution can induce distinct kinetic models, even under identical physiological conditions.

## Supporting information

**S1 Fig. The monomeric INSv that produced by the refolding process. (A)** 20% SDS-PAGE analysis of INSv, Marker 1 indicates proinsulin (Iodinated Human proinsulin, Sigma-Aldrich, 9015) (~12 KDa), and Marker 2 indicates monomeric insulin aspart (NovoRapid™, Novo Nordisk, Denmark) (~5 kDa). Soluble INSv contains non-purified proINSv (12 KDa) and digested INSv (5 kDa), which is colored yellow. Purified and concentrated proINSv have been highlighted in red. INSv is performed enzymatic digestion with trypsin through zero time ($h_0$) and 4-hour ($h_4$) at 30°C, which are indicated by green arrows. Each line consists of 2.5 mg/ml pro- and mature INSv samples. **(B)** Circular Dichroism plot of distinct buffered monomeric INSv at pH 3, pH 10, and pH 9.5.
(TIF)

**S2 Fig. Statistical analysis of the AUC measurement.** Analysis indicated that INSv application significantly changed spontaneous calcium activity (IAsp *p*-value: 0.1290, INSv *p*-value: 0.0002). The mean AUC decreased in both samples.
(TIF)

**S3 Fig. Selected segments from various recordings of calcium activity.** It is associated with IAsp **(A)** and INSv **(B)**. For both panels, **a)** a pre-stimulation network image and **b-d)** a progressive spread of calcium activity across the network
(TIF)

**S4 Fig. Hexamer INSv data collection and structure determination process. (A)** INSv hexamers (20 um) has been crystallized in 2.4 M NaCl, 100 mM Tris-HCl at pH 7.4, 6 mM ZnCI2, 20 (w/v) poly(ethylene glycol) PEG-8000 buffer. **(B)** Multiple data collection has been performed by Turkish Light Source at ambient temperature. Main hardware components of Turkish Light Source. (1) X-ray source, (2) four-circle Kappa goniometer, (3) shutter, (4) collimator, (5) beamstop, (6) X-ray detector, (7) video microscope, (8) low-temperature insert, (9) XtalCheck module. (*adapted from Atalay et al., 2023; Gul et al., 2023*). **(C)** Double mutated INSv hexamer has been determined as a dimer in the asymmetric unit at 2.5 A resolution. 2*Fo-Fc* simulated annealing-omit map at 1 sigma level is colored in slate
(TIF)

**S5 Fig. Projection of the peptide-specific deuterium exchange. (A)** Relative deuterium exchange of INSv and IAsp peptides depicted by Woods plots over the 200 min time course. **(B)** Comparison between time-dependent relative deuterium exchange of representative INSv and IAsp peptides. Data is represented on a logarithmic scale.
(TIF)

**S6 Fig. Projection of the peptide-specific deuterium exchange.** Relative deuterium exchange of INSv and IAsp peptides depicted by Woods plots over the 12-sec, 120-sec, 20-min, and 24-hour time courses
(TIF)

**S7 Fig. Butterfly plot for deuterium incorporation of INSv (LysB22-AspB28) and IAsp (AspB28).** Apart from overall dynamic variability, between 20–25 residues in both conformers indicates substantial mobility over all time intervals
(TIF)

**S8 Fig. Detailed projection of the peptide-specific deuterium exchange. (A)** Relative deuterium exchange map of INSv and IAsp peptides over the all-time courses. **(B)** Comparison between time-dependent relative deuterium exchange of representative INSv and IAsp peptides. Data is represented on a logarithmic scale. X-axes indicate relative uptake (Da), and Y-axes demonstrate exposure time (min). The black arrow shows the mutated residue in the INSv map.
(TIF)

**S1 Table. A summary table providing key information about the HDX-MS data.**
(TIF)

## Acknowledgement

The authors gratefully acknowledge the use of the services and facilities of the Istanbul Technical University MOBGAM (Molecular Biology-Biotechnology&Genetics Research Center), University of Health Science-Validebag DETAUM (Experimental Medicine Research and Application Center), Istanbul Medipol University, Research Institute for Health Sciences and Technologies (SABITA).

## Author contributions

**Conceptualization:** Esra Ayan, Gizem Dinler-Doğanay, Hasan DeMirci.

**Data curation:** Esra Ayan, Miray Türk, Özge Tatlı, Sevginur Bostan, Elek Telek.

**Formal analysis:** Esra Ayan, Miray Türk, Sevginur Bostan.

**Funding acquisition:** Hasan DeMirci.

**Investigation:** Esra Ayan.

**Methodology:** Esra Ayan, Miray Türk, Sevginur Bostan, Elek Telek, Baran Dingiloğlu.

**Project administration:** Esra Ayan, Gizem Dinler-Doğanay, Hasan DeMirci.

**Resources:** Ahmet Katı, Gizem Dinler-Doğanay, Hasan DeMirci, Muhammed İkbal Alp.

**Software:** Esra Ayan, Miray Türk, Özge Tatlı, Sevginur Bostan, Elek Telek, Baran Dingiloğlu.

**Supervision:** Esra Ayan, Gizem Dinler-Doğanay, Hasan DeMirci.

**Validation:** Esra Ayan, Özge Tatlı, Sevginur Bostan, Elek Telek, B. Züleyha Doğan.

**Visualization:** Esra Ayan.

**Writing – original draft:** Esra Ayan.

**Writing – review & editing:** Esra Ayan, Miray Türk, Hasan DeMirci.

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
