## [Decision Letter · Decision Letter 0]

14 Nov 2024

PONE-D-24-41868X-ray Crystallographic and Hydrogen Deuterium Exchange Studies Confirm Alternate Kinetic Models for Homolog Insulin MonomersPLOS ONE

Dear Dr. Ayan,

Thank you for submitting your manuscript to PLOS ONE. After careful consideration, we feel that it has merit but does not fully meet PLOS ONE’s publication criteria as it currently stands. Therefore, we invite you to submit a revised version of the manuscript that addresses the points raised during the review process.

We look forward to receiving your revised manuscript.

Kind regards,

Elias John Fernandez, Ph.D.

Academic Editor

PLOS ONE

Journal Requirements:

3. Thank you for stating the following financial disclosure: [E.T. was supported by the University of Pécs Medical School, a grant from Dr. Szolcsányi János Research Fund (KA-2022-09), as well as the grant of Dr. Romhányi György fellowship for young scientists (ÁOK-IK). This research was financially supported by The Scientific and Technological Research Council of Turkey (TUBITAK) (Project no: 122R061). E.A. was supported by received funding from the TUBITAK 2244 Program (Project no: 119C132). Additionally, M.T. and B.D. received funding from the TUBITAK 2211 Program.]. Please state what role the funders took in the study. If the funders had no role, please state: "The funders had no role in study design, data collection and analysis, decision to publish, or preparation of the manuscript." If this statement is not correct you must amend it as needed. Please include this amended Role of Funder statement in your cover letter; we will change the online submission form on your behalf.

5. Please include captions for your Supporting Information files at the end of your manuscript, and update any in-text citations to match accordingly. Please see our Supporting Information guidelines for more information: http://journals.plos.org/plosone/s/supporting-information .

6. PLOS ONE now requires that authors provide the original uncropped and unadjusted images underlying all blot or gel results reported in a submission’s figures or Supporting Information files. This policy and the journal’s other requirements for blot/gel reporting and figure preparation are described in detail at https://journals.plos.org/plosone/s/figures#loc-blot-and-gel-reporting-requirements and https://journals.plos.org/plosone/s/figures#loc-preparing-figures-from-image-files. When you submit your revised manuscript, please ensure that your figures adhere fully to these guidelines and provide the original underlying images for all blot or gel data reported in your submission. See the following link for instructions on providing the original image data: https://journals.plos.org/plosone/s/figures#loc-original-images-for-blots-and-gels. In your cover letter, please note whether your blot/gel image data are in Supporting Information or posted at a public data repository, provide the repository URL if relevant, and provide specific details as to which raw blot/gel images, if any, are not available. Email us at plosone@plos.org if you have any questions.

Reviewers' comments:

Reviewer's Responses to Questions

**Comments to the Author**

1. Is the manuscript technically sound, and do the data support the conclusions?

Reviewer #1: Yes

Reviewer #2: Partly

2. Has the statistical analysis been performed appropriately and rigorously? 

Reviewer #1: N/A

Reviewer #2: I Don't Know

3. Have the authors made all data underlying the findings in their manuscript fully available?

Reviewer #1: Yes

Reviewer #2: Yes

4. Is the manuscript presented in an intelligible fashion and written in standard English?

Reviewer #1: Yes

Reviewer #2: No

5. Review Comments to the Author

Reviewer #1: The manuscript by Ayan et al. describes a combined biophysical and crystallographic study of INSv – a double mutant of insulin. IAsp insulin containing a single PB28D mutation in the B chain was chosen for comparison. INSv contained an additional RB22K mutation. The lower stability of INSv compared to IAsp was interrogated using DSC, HDX-MS, computational GNM analysis, and analysis of crystallographic B factors. It was concluded that R-to-K substitution in the B22 position may significantly change the protein dynamics, including inducing distinct kinetic models. The results convincingly showed that INSv is indeed more dynamic than IAsp, with the HDX data correlating with the B-factor analysis from the crystal structure obtained at ambient temperature.

The only major request to the authors is to elaborate about the following points in the Introduction. The reasoning of why the specific mutation RB22K was chosen needs to be given. It becomes slightly clearer later in the paper that position B22 is functionally important, but this has to be stated and described in the Intro. Also, to the same point, the premise of the study is unclear. If INSv had been described previously, that has to be stated and explained; if not, then the authors need to elaborate on how they came up with this insulin design.

Throughout the manuscript the insulin’s oligomeric state is described as monomeric, dimeric, hexameric, creating quite a confusion. For example, the crystal structures of INSv and IAsp are hexameric. Then, which measurements provided evidence that the insulins are monomeric in solution?

INSv crystal structure was obtained at ambient T, but it is compared directly with the 100K structure of IAsp (4GBN). Also, the pHs were different for the crystal growths of these proteins, 6.5 for 4GBN and 7.4 for INSv. These differences in the experimental conditions needs to be taken into account, acknowledged and clearly stated in the manuscript.

The title for HDX-MS in the Methods states “Higher-order insulin monomers”. Is that even possible? Perhaps, oligomers rather?

Crystallization was carried out under oil, but it is called vapor diffusion micro-batch. If crystallization is under oil. it is not vapor diffusion.

All the references at the end of the manuscript are cited as “Epub ahead of print” even those published 20 years ago. Correct all the references.

Reviewer #2: Authors are strongly recommended to use a language-editing service to improve the manuscript.

The manuscript presents complementary datasets that show unique solution properties of the INSv variant relative to IAsp. The data presented has potential to be invaluable to the field in general. However, additional studies are necessary to explain the relationship between cellular activities and the solution dynamics of the insulin species presented in the manuscript. A major concern is evidence (or lack thereof) of the monomeric state. The authors aim to characterize solution dynamics of the monomeric form of insulin. As such, it is critical to show that the current form under study is, in fact, monomeric. The authors briefly allude to preparation of the monomeric form, but did not present significant data indicating this. Another source of confusion is that the monomeric form of insulin is roughly 5,000 Daltons, but the authors used a 10K concentrator to buffer exchange.

Organization of the paper can be significantly improved. The sub-headings in the results section sometimes do not match exactly what is covered in that sub-section. The authors used the newly determined crystal structure in their GNM analysis. It might be better, then, to move the discussion of the crystal structure ahead of any GNM analysis to improve the organization and flow of the paper.

Authors are recommended to include a discussion of statistical tests and analyses used in the methods section.

Figure 2. Authors depict insulin as a single chain in panel B, consistent with their analyses and as discussed in a previous section. They are recommended to indicate Chain A and Chain B in the figure anyway to improve readability, and matching panel A.

Authors need to define what hinge regions are and how they are determined. The methods section corresponding to the GNM analysis is not comprehensive enough to guide the reader in arriving at their conclusions. Consistency between the GNM analyses and HDX data is not apparent in Figure 2, specifically looking at the panel labelled as kinetically hot regions. The same can be said about the INSv analysis.

While it is clear from the data presented that there is significantly higher exchange in INSv compared to IAsp, it is not clear how this trend is precisely related to results from DSC. There is a complex relationship between the change in entropy, local/global dynamics, structure, and shape of proteins. The analysis presented seem to oversimplify this.

Figure 4 might be better as a supplementary figure. Consistency in labeling is also recommended (IAsp vs. Aspart).

In Figure 5, it might be better to omit the word ‘spectrum’ in referring to the coloring of the figure.

In Page 28, there is a stray sentence at the end of the first paragraph.

Does PyMol need to be cited in every figure or instance it is used? Perhaps a section can be added in the methods indicating that all figures were generated using PyMol. The relevant PyMol citations can then be referenced.

In Figure 7, the author needs to state how the amino acid residues are colored. It seems the color matches the presented model, that is, colored by experimental b-factor. Nonetheless, this needs to be explicitly stated.

Perhaps ‘solution structure’ can be used instead of ‘in-solution structure?’

6. PLOS authors have the option to publish the peer review history of their article (what does this mean? ). If published, this will include your full peer review and any attached files.

**Do you want your identity to be public for this peer review?** For information about this choice, including consent withdrawal, please see our Privacy Policy .

Reviewer #1: **Yes: ** Andrey Kovalevsky

Reviewer #2: No

---

## [Author Response · Author response to Decision Letter 1]

27 Dec 2024

Reviewer #1:

Comment 1. The manuscript by Ayan et al. describes a combined biophysical and crystallographic study of INSv – a double mutant of insulin. IAsp insulin containing a single PB28D mutation in the B chain was chosen for comparison. INSv contained an additional RB22K mutation. The lower stability of INSv compared to IAsp was interrogated using DSC, HDX-MS, computational GNM analysis, and analysis of crystallographic B factors. It was concluded that R-to-K substitution in the B22 position may significantly change the protein dynamics, including inducing distinct kinetic models. The results convincingly showed that INSv is indeed more dynamic than IAsp, with the HDX data correlating with the B-factor analysis from the crystal structure obtained at ambient temperature.

Response 1. We sincerely thank our reviewer for their thoughtful comments on this study.

Comment 2. The only major request to the authors is to elaborate about the following points in the Introduction. The reasoning of why the specific mutation RB22K was chosen needs to be given. It becomes slightly clearer later in the paper that position B22 is functionally important, but this has to be stated and described in the Intro. Also, to the same point, the premise of the study is unclear. If INSv had been described previously, that has to be stated and explained; if not, then the authors need to elaborate on how they came up with this insulin design.

Response 2. We thank the reviewer for their constructive comments. The RB22K mutation in insulin was specifically designed to address both biochemical and industrial challenges associated with insulin production and functionality. Following, we elaborate on the rationale for this mutation and its implications. (i) the ArgB22 to Lys mutation was chosen to mitigate undesired cleavage products during the production process, as trypsin exhibits significantly higher cleavage specificity for Arg compared to Lys. By substituting Arg with Lys, the risk of proteolytic degradation is reduced, enhancing product consistency and yield. (ii) it has been well-established that Lys, compared to Arg, introduces greater structural flexibility to proteins due to its shorter and less bulky side chain. This increased flexibility can influence the dynamic stability of insulin, a key factor in both its functional behavior and receptor binding. (iii) while this increased flexibility might reduce stability in certain contexts, it also makes the protein more adaptable to industrial processes, such as formulation and delivery system optimization. (iv) this mutation simplifies the manufacturing process by allowing insulin to be processed with trypsin alone, eliminating the need for additional enzymes like carboxypeptidase. This not only reduces production costs but also streamlines the overall process, aligning with industrial efficiency goals. The result is a time-efficient and scalable production pipeline, yielding a highly effective insulin analog. (v) position B22 in insulin is functionally critical due to its role in maintaining structural integrity and facilitating receptor binding. Studies have shown that mutations in this region can significantly alter the dynamic behavior and stability of insulin. The RB22K mutation provides a unique opportunity to investigate how subtle changes in side-chain properties influence insulin’s structural dynamics and functional outcomes. (vii) replacing Arg with Lys lowers the isoelectric point (pI) of the insulin product, which facilitates the dissociation of the hexameric (inactive) form into the monomeric (active) form under physiological conditions. This is crucial for therapeutic efficacy as the monomeric form is the biologically active state. To the best of our knowledge, INSv has not been previously described in the literature. This study represents a novel exploration of the RB22K mutation, combining crystallographic and biophysical methods to provide new insights into insulin’s structure-function relationships and its potential industrial applications.

Accordingly, we added below text into the introduction:

“The ArgB22 residue in insulin was specifically mutated to Lys (RB22K) to address both biochemical and industrial challenges. This mutation reduces the formation of undesired cleavage products, as trypsin has a higher specificity for Arg than Lys [20], thereby improving production consistency and yield. Additionally, replacing Arg with Lys lowers the isoelectric point (pI) of the insulin molecule, promoting the dissociation of the inactive hexameric form into the biologically active monomer under physiological conditions [21]. Lys also introduces greater flexibility to the insulin structure due to its shorter side chain, which can influence the stability and receptor-binding dynamics of the protein [22]. Importantly, this mutation eliminates the need for additional enzymes, such as carboxypeptidase [23], allowing insulin to be processed using trypsin alone, streamlining production and reducing costs. The RB22K mutation not only addresses industrial optimization goals but also provides a valuable model for studying the structural and functional implications of modifications at the functionally critical B22 position.”

Comment 3. Throughout the manuscript the insulin’s oligomeric state is described as monomeric, dimeric, hexameric, creating quite a confusion. For example, the crystal structures of INSv and IAsp are hexameric. Then, which measurements provided evidence that the insulins are monomeric in solution?

Response 3. Thank you for your insightful comments and constructive feedback. Below, we address your concerns regarding the monomeric state of the insulin variant under study. We acknowledge the importance of verifying the monomeric state of the insulin molecules in our study.

• Prior studies have demonstrated that insulin undergoes monomerization at alkaline pH values. Specifically, Tokumoto et al. (2006) showed that monomerization is promoted at high pH levels (e.g., pH 10) compared to neutral pH.

• Similarly, Gursky et al. (1992) confirmed through structural biology analyses that insulin remains in a monomeric state at pH 9, 10, and 11 without major conformational changes.

• Furthermore, a biochemical study has shown that insulin dissociation begins at pH 8 and is completed by pH 10, providing additional support for monomerization under these conditions.

• In our own work, intact mass spectrometry (MALDI-TOF) analysis demonstrated that both IAsp and INSv variants are monomeric in solution at pH 9. The molecular weights of IAsp (~5977 Da) and INSv (~5823 Da) confirm their monomeric states (Figure A). Although these data are of publishable quality, we opted not to include the figures directly in the manuscript at this stage, but the added references into the text.

[Figure A. Comparative MALDI-TOF intact mass analysis between IAsp and INSv monomers.]

in manuscript: “Insulin is primarily hexameric (consist of six identical monomer ~36 KDa) and dimeric (consist of two identical monomer ~12 KDa) during production, delivery, and circulation. However, upon binding to its receptor, the protein adopts a monomeric conformation (~5 KDa) [2], which differs from its solution structure observed in NMR and X-ray crystallographic studies [3,4].”

“Precipitation through pH adjustment and subsequent solubilization in 25 mM Tris-HCl pH 9.0 buffer were employed to convert IAsp (NovoRapid) from a hexameric to a monomeric state [11,12, 27]. For further buffer exchange and concentration of INSv monomer, a 3K amicon ultrafiltration unit was used ten times washing with 25 mM Tris-HCl pH 9.0 buffer.”

Comment 4. INSv crystal structure was obtained at ambient T, but it is compared directly with the 100K structure of IAsp (4GBN). Also, the pHs were different for the crystal growths of these proteins, 6.5 for 4GBN and 7.4 for INSv. These differences in the experimental conditions needs to be taken into account, acknowledged and clearly stated in the manuscript.

Response 4. We thank our reviewer for highlighting the importance of addressing differences in experimental conditions between the INSv and IAsp crystal structures. We have revised the manuscript to explicitly acknowledge these differences. Specifically, we have added the following sentence to the manuscript:

in manuscript: "For comparison, the IAsp structure, determined under cryogenic conditions at pH 6.5, was used as the reference model, whereas INSv crystals were grown at pH 7.4."

Additionally, we have included a text on how temperature differences during data collection might influence structural observations, particularly the outward orientation of Asn3 in INSv compared to IAsp:

"… This outward orientation of Asn3 of INSv compared to IAsp may also be due to the temperature differences during the data collection, but thermal displacement (b-factor) analysis and computational GNM analyses of the INSv and IAsp crystal structures closely resemble the dynamics observed in insolution analyses of monomer INSv and IAsp (such as HDX-MS, DSC)."

These revisions acknowledge and account for the differences in pH and temperature between the two crystal growth conditions and data collection methods. Furthermore, we emphasize that the structural dynamics derived from the crystal structures are consistent with solution-based experimental techniques, supporting the validity of the comparison. We hope these clarifications address the reviewer’s concerns.

Comment 5. The title for HDX-MS in the Methods states “Higher-order insulin monomers”. Is that even possible? Perhaps, oligomers rather?

Response 5. We thank our reviewer for their insightful comment and fully agree with this observation. To clarify, we determined that our samples were monomers through MALDI-TOF intact mass analysis. Additionally, as supported by the literature, insulin predominantly exists as a monomer at pH values above 8 (Figures and references provided in Comment 3 further illustrate this point). However, we greatly acknowledge the potential for confusion and have addressed this by adopting a more appropriate and precise title for the manuscript: “Kinetics of insulin monomers defined through HDX-MS analysis” We hope this revision resolves our reviewer’s concerns.

Comment 6. Crystallization was carried out under oil, but it is called vapor diffusion micro-batch. If crystallization is under oil. it is not vapor diffusion.

Response 6. We apologize for this inconvenience. We edited this mistake adding the “sitting drop, microbatch under oil method”

Comment 7. All the references at the end of the manuscript are cited as “Epub ahead of print” even those published 20 years ago. Correct all the references.

Response 7. Thank our reviewer for this comment. We edited all references accordingly.

Reviewer #2:

Comment 1. Authors are strongly recommended to use a language-editing service to improve the manuscript.

Response 1. Thank you for your valuable suggestion regarding language editing. I have carefully revised the manuscript and utilized a professional language-editing service to enhance its clarity and readability.

I hope that these improvements address your concerns.

Comment 2. The manuscript presents complementary datasets that show unique solution properties of the INSv variant relative to IAsp. The data presented has potential to be invaluable to the field in general. However, additional studies are necessary to explain the relationship between cellular activities and the solution dynamics of the insulin species presented in the manuscript. A major concern is evidence (or lack thereof) of the monomeric state. The authors aim to characterize solution dynamics of the monomeric form of insulin. As such, it is critical to show that the current form under study is, in fact, monomeric. The authors briefly allude to preparation of the monomeric form but did not present significant data indicating this. Another source of confusion is that the monomeric form of insulin is roughly 5,000 Daltons, but the authors used a 10K concentrator to buffer exchange.

Response 2. Thank you for your insightful comments and constructive feedback. We deeply appreciate your detailed evaluation of our manuscript and the valuable points you raised. Below, we address your concerns regarding the monomeric state of the insulin variant under study and the use of a 10K concentrator. We acknowledge the importance of verifying the monomeric state of the insulin molecules in our study.

• Prior studies have demonstrated that insulin undergoes monomerization at alkaline pH values. Specifically, Tokumoto et al. (2006) showed that monomerization is promoted at high pH levels (e.g., pH 10) compared to neutral pH.

• Similarly, Gursky et al. (1992) confirmed through structural biology analyses that insulin remains in a monomeric state at pH 9, 10, and 11 without major conformational changes.

• Furthermore, a biochemical study has shown that insulin dissociation begins at pH 8 and is completed by pH 10, providing additional support for monomerization under these conditions.

• In our own work, intact mass spectrometry (MALDI-TOF) analysis demonstrated that both IAsp and INSv variants are monomeric in solution at pH 9. The molecular weights of IAsp (~5977 Da) and INSv (~5823 Da) confirm their monomeric states (Figure A). Although these data are of publishable quality, we opted not to include the figures directly in the manuscript at this stage, but the revised text now describes these findings in detail

[Figure A. Comparative MALDI-TOF intact mass analysis between IAsp and INSv monomers.]

We sincerely apologize for the oversight and any confusion caused regarding the use of a 10K concentrator. This was a miscommunication on our part. To clarify: The 10K concentrator was used during buffer exchange for preproinsulin (~10 kDa). However, for mature monomeric insulin, we typically employed pH precipitation methods then a 3K Amicon filter. This correction has been made in the main text as below, and the revised version explicitly states the appropriate methods used for the concentration and preparation of the insulin monomer:

“For further buffer exchange and concentration of INSv monomer, a 3K amicon ultrafiltration unit was used ten times washing with 25 mM Tris-HCl pH 9.0 buffer.”

Comment 3. Organization of the paper can be significantly improved. The sub-headings in the results section sometimes do not match exactly what is covered in that sub-section. The authors used the newly determined crystal structure in their GNM analysis. It might be better, then, to move the discussion of the crystal structure ahead of any GNM analysis to improve the organization and flow of the paper.

Response 3. Thank you for your thoughtful and constructive feedback. We have carefully revised the manuscript to address the organizational issues you highlighted. Specifically: (i) inconsistent subheadings have been corrected to ensure alignment with the content of their respective sub-sections. The first sub-section has been divided for better clarity and focus, (ii) to enhance the logical flow of the manuscript, the discussion of the newly determined crystal structure has been moved ahead of the experimental H/D exchange kinetics and computational GNM dynamics sections. We believe these changes significantly improve the organization and readability of the results section. Thank you again for helping us refine our work.

Comment 4. Authors are recommended to include a discussion of statistical tests and analyses used in the methods section.

Response 4. Thank you very much for your feedback. We added statistical tests and analyses used in the manuscript as below:

“Deuterium uptake data report was obtained from DynamX Software after HDX-MS analysis and obtained time-dependent Relative Fractional Uptake (RFU) values were presented as XY graphs generated by GraphPad Prism 8.1.Imaging was conducted using 10x objectives. Images were captured at a frequency of 1 frame per second and subsequently compiled into video recordings. These recordings were analyzed using Zen Blue (Zeiss) software, where cell bodies were designated as regions of interest (ROIs). Calcium transient intensities were determined using

---

## [Decision Letter · Decision Letter 1]

30 Jan 2025

X-ray crystallographic and hydrogen deuterium exchange studies confirm alternate kinetic models for homolog insulin monomers

PONE-D-24-41868R1

Dear Dr. Ayan,

We’re pleased to inform you that your manuscript has been judged scientifically suitable for publication and will be formally accepted for publication once it meets all outstanding technical requirements.

Kind regards,

Elias John Fernandez, Ph.D.

Academic Editor

PLOS ONE

Additional Editor Comments (optional):

Reviewers' comments:

Reviewer's Responses to Questions

**Comments to the Author**

1. If the authors have adequately addressed your comments raised in a previous round of review and you feel that this manuscript is now acceptable for publication, you may indicate that here to bypass the “Comments to the Author” section, enter your conflict of interest statement in the “Confidential to Editor” section, and submit your "Accept" recommendation.

Reviewer #1: All comments have been addressed

Reviewer #2: All comments have been addressed

2. Is the manuscript technically sound, and do the data support the conclusions?

Reviewer #1: (No Response)

Reviewer #2: Yes

3. Has the statistical analysis been performed appropriately and rigorously? 

Reviewer #1: N/A

Reviewer #2: Yes

4. Have the authors made all data underlying the findings in their manuscript fully available?

Reviewer #1: Yes

Reviewer #2: Yes

5. Is the manuscript presented in an intelligible fashion and written in standard English?

Reviewer #1: Yes

Reviewer #2: Yes

6. Review Comments to the Author

Reviewer #1: (No Response)

Reviewer #2: The authors have addressed all previous comments. The organization and flow of the paper is much improved.

7. PLOS authors have the option to publish the peer review history of their article (what does this mean? ). If published, this will include your full peer review and any attached files.

**Do you want your identity to be public for this peer review?** For information about this choice, including consent withdrawal, please see our Privacy Policy .

Reviewer #1: **Yes: ** Andrey Kovalevsky

Reviewer #2: No

---

## [Editor Report · Acceptance letter]

PONE-D-24-41868R1

PLOS ONE

Dear Dr. Alp,

I'm pleased to inform you that your manuscript has been deemed suitable for publication in PLOS ONE. Congratulations! Your manuscript is now being handed over to our production team.

Kind regards,

on behalf of

Dr. Elias John Fernandez

Academic Editor

PLOS ONE